# Semi-TSGAN: Semi-Supervised Learning for Highlight Removal Based on Teacher-Student Generative Adversarial Network

**DOI:** 10.3390/s24103090

**Published:** 2024-05-13

**Authors:** Yuanfeng Zheng, Yuchen Yan, Hao Jiang

**Affiliations:** School of Electronic Information, Wuhan University, Wuhan 430072, China; zhengyuanfeng@whu.edu.cn (Y.Z.); yyc@whu.edu.cn (Y.Y.)

**Keywords:** semi-supervised learning, highlight removal, generative adversarial network

## Abstract

Despite recent notable advancements in highlight image restoration techniques, the dearth of annotated data and the lightweight deployment of highlight removal networks pose significant impediments to further advancements in the field. In this paper, to the best of our knowledge, we first propose a semi-supervised learning paradigm for highlight removal, merging the fusion version of a teacher–student model and a generative adversarial network, featuring a lightweight network architecture. Initially, we establish a dependable repository to house optimal predictions as pseudo ground truth through empirical analyses guided by the most reliable No-Reference Image Quality Assessment (NR-IQA) method. This method serves to assess rigorously the quality of model predictions. Subsequently, addressing concerns regarding confirmation bias, we integrate contrastive regularization into the framework to curtail the risk of overfitting on inaccurate labels. Finally, we introduce a comprehensive feature aggregation module and an extensive attention mechanism within the generative network, considering a balance between network performance and computational efficiency. Our experimental evaluations encompass comprehensive assessments on both full-reference and non-reference highlight benchmarks. The results demonstrate conclusively the substantive quantitative and qualitative enhancements achieved by our proposed algorithm in comparison to state-of-the-art methodologies.

## 1. Introduction

Specular highlights, a common optical phenomenon observed in the real world, manifest as bright spots on glossy surfaces when illuminated [1]. The removal of highlights has long been a foundational challenge in computer vision. As Figure 1 depicts, these highlights are pivotal for enhancing the performance of various vision tasks, including object detection [2], intrinsic image decomposition [3], and tracking [4]. For the sake of simplicity, it is important to note that we refer to highlights as specular highlights in this paper unless otherwise specified. In recent times, numerous deep learning-based methods [5,6,7,8] have emerged to tackle image restoration problems. Within the specific domain of highlight removal, substantial efforts have been dedicated. In contrast to traditional methods heavily reliant on manually designed priors, deep learning-based solutions demonstrate superior restoration outcomes owing to their data-driven nature.

However, the exceptional efficacy of data-driven deep learning highlight removal algorithms often hinges upon supervised learning utilizing high-quality paired datasets. It is widely acknowledged that amassing substantial quantities of paired image featuring highlights and highlight-free images demands extensive human resources, material investment, and financial commitment, further complicated by stringent and challenging acquisition conditions. One potential approach involves acquiring a limited set of paired data supplemented by unpaired instances to augment algorithmic performance, striking a delicate balance between algorithmic efficacy and economic feasibility. Precisely, the challenge of this scenario with wider application prospects is endeavored to be tackled in this paper. Our scenario entails partially robust supervised signals alongside a considerable volume of unsupervised signals. Leveraging semi-supervised learning methodologies enables the simultaneous utilization of annotated and unannotated data during model training. Typically, differentiated loss functions are employed in semi-supervised training for these two data categories. Supervised learning’s analogous loss function is applied to annotated data, while techniques such as consistency regularization [9], transfer learning [10], and few-shot learning [11] are customary for unannotated data, aiming to maximize dataset utility. To be specific, our methodology is grounded in the mean-teacher approach [12]. This method provides a mechanism for generating pseudo labels for unlabeled data and employs a consistency loss function to augment network accuracy and resilience. In essence, it constructs a teacher model, refined through an exponential moving average (EMA) strategy from a student model with enhanced performance. The teacher model’s predictions act as pseudo labels to guide the training of the student model. However, tailoring the mean-teacher method to address the challenges inherent in highlight removal is a non-trivial endeavor. Several reasons contribute to this complexity: (1) there exists no assurance that the teacher model consistently outperforms the student model—erroneous pseudo labels may significantly compromise the training of the student network; (2) the prevalent use of the consistency loss relying on L1 distance presents a challenge—the stringent nature of the “strict” L1 loss function may predispose the model to overfitting erroneous predictions, thereby leading to confirmation bias. In conjunction with the limitations observed in the mean-teacher model framework, the prevalent GAN-based highlight removal model is often excessively intricate, demanding considerable storage and computational resources. This complexity poses challenges in terms of practical deployment, as it hinders operational efficiency and is not conducive to streamlined deployment processes.

To ameliorate the initial concern, we establish a dependable repository to archive optimal teacher outputs, serving as pseudo labels. The primary challenge lies in identifying the criteria defining these “best-ever” outputs. While non-reference image quality assessment (NR-IQA) appears intuitive for this purpose, existing metrics for highlight images, as delineated in [13,14,15], display certain disparities with human visual perception. To discern the most suitable metric for our objectives, we systematically compare several NR-IQA metrics, employing the monotonicity property as a reliability criterion. Empirical analysis indicates that MUSIQ [16] aligns best with this criterion. Regarding the secondary issue, we introduce contrastive learning as an ancillary regularization approach to mitigate overfitting. Diverging from conventional loss functions primarily assessing output-ground truth proximity, contrastive loss introduces supplementary guidance to forestall output degradation. Within our semi-supervised learning framework, wherein access is limited to degraded images from the unlabeled dataset, contrastive regularization proves particularly advantageous. It facilitates the model’s utilization of unlabeled data, enhancing its learning efficiency. Considering the balance of network performance and model size, we design and generate the network with the following key features: (1) introduce a full-scale feature aggregation module, and first adjustment of the feature dimension through point-wise convolution and then, the depth convolution of different sizes is used to extract more characteristics of different scale and learn more comprehensive semantic information, which helps comprehensively improve the highlight suppression effect; (2) focus on the highlights area, we design the full-scale aggregation module to comprehensively tap the characteristics of key areas; (3) we learn end-to-end highlight to highlight-free direct mapping without sub-network—the training process is simple, and the parameters are small. Combined with full-scale attention mechanism, the design can tap key areas in a targeted manner to prevent the network from consuming too much computing resources in other areas, thereby achieving precise feature extraction of high-efficiency highlight removal. Our proposed methodologies are substantiated through extensive experimental validations, providing empirical evidence that attests to the efficacy and robustness of our design considerations.

In summary, our primary contributions can be summarized as follows:

To the best of our knowledge, we are the first ones to introduce a semi-supervised highlight removal framework named Semi-TSGAN, based on the mean-teacher approach. This framework effectively capitalizes on unlabeled data to enhance the model’s adaptability and performance when applied to real-world data scenarios.

We systematically evaluate teacher outputs using a meticulously selected No-Reference Image Quality Assessment (NR-IQA) metric and establish a dependable repository for storing optimal teacher outputs. This process ensures the reliability and credibility of the generated pseudo-labels.

We employ contrastive loss as a means of regularization to mitigate confirmation bias, thereby enhancing the model’s ability to learn and generalize effectively.

We introduce a comprehensive feature aggregation module and an extensive attention mechanism, considering both network performance and computational efficiency.

## 2. Related Work

### 2.1. Traditional Approaches

Traditional model-based highlight removal methods are typically classified into two categories: multiple-image and single-image approaches. Multiple-image methods often leverage viewpoint dependence to identify matching specular and diffuse pixels across several images [17,18,19]. While these techniques yield commendable results, they are often computationally intensive. On the other hand, addressing single-image highlight removal poses a more complex and longstanding challenge. In this domain, conventional methods rely on prior knowledge, albeit the simplistic assumptions frequently fall short when applied to real-world images. Guo et al. introduced a sparse and low-rank reflection model for detecting and removing specular highlights from a single input image [20]. Shen et al. undertook the separation of specular highlight reflections in a color image, employing error analysis of chromaticity and judicious selection of body color for each pixel [21]. Subsequent work by Shen and Cai extended these efforts to enhance algorithmic robustness [22]. For natural images, effective specular highlight removal is achieved through the application of the dichromatic reflection model. Akashi et al. proposed a modified variant of sparse non-negative matrix factorization (NMF) without spatial priors [23]. Shen and Zheng adopted a color space approach for analyzing the diffuse and specular component distributions, leveraging this insight for effective separation [24]. Yang et al. introduced a method to disentangle the diffuse and specular reflection components within the Hue, Saturation, and Intensity (HSI) color space, emphasizing its applicability in real-time scenarios [25]. Tan et al. devised an interactive approach, treating specular highlight removal as an inpainting process [26]. While methods based on prior knowledge demonstrate efficacy, they often fall short in unconstrained environments. The intrinsic ill-posed nature of the problem necessitates the utilization of prior knowledge or assumptions about the characteristics of natural images to render the problem tractable. The efficacy of these approaches hinges on the accuracy of estimated geometry, illumination, reflectance, and material properties. Notably, existing traditional algorithms struggle to semantically disambiguate highlights and pure white in intricate real-world scenarios.

### 2.2. Deep Learning Based Approaches

Recent strides in early deep learning-based highlight removal [27,28] predominantly revolve around harnessing physical imaging models. Specifically, these approaches leverage neural networks to estimate both diffusion and specular components. Funke et al. introduced a GAN-based method designed for the automatic removal of specular highlights from individual endoscopic images [29]. Training this network involves extracting small image patches with specular highlights and patches without highlights from endoscopic videos. Wu et al. introduced an innovative Generative Adversarial Network (GAN) for specular highlight removal grounded in polarization theory [30]. Lin et al. proposed a novel learning approach in the form of a fully convolutional neural network (CNN), systematically eradicating specular highlights from single images by generating their diffuse components [28]. To enhance generalization, they curated a synthetic dataset. Muhammad et al. subsequently presented Spec-Net and Spec-CGAN, tailored for eliminating high-intensity specularity from low chromaticity facial images [8]. Fu et al. devised a multi-task network designed for simultaneous highlight detection and removal [5], leveraging a sizable dataset with meticulous manual annotations. However, these methods heavily depend on supervised learning, relying on extensive training data to develop a robust model. A recurrent issue in current research findings pertains to the complexity of deep neural networks, characterized by intricate structures and a considerable number of parameters. This complexity poses challenges for practical deployment, especially in resource-constrained applications such as mobile or embedded devices. Consequently, there is a pressing need for solutions that advocate lightweight neural networks, aligning with the evolving requirements of diverse applications.

### 2.3. Semi-Supervised Learning

Semi-supervised Learning is a learning paradigm associated with constructing models that use both labeled and unlabeled data. SSL methods can improve learning performance by using additional unlabeled instances compared to supervised learning algorithms, which can use only labeled data. It is easy to obtain SSL algorithms by extending supervised learning algorithms or unsupervised learning algorithms. Many semi-supervised methods have been developed, such as mean teacher [12], transfer learning [10], and few-shot learning [11]. This success also triggers its applications to other vision tasks [31]. Unfortunately, to the best of our knowledge, semi-supervised learning has been rarely explored in highlight removal image. Among them, the mean-teacher method, consisting of two models called Student and Teacher, which is based on consistency regularization, has achieved remarkable success in semi- supervised image recognition. Mean teacher averages model weights using EMA over training steps and tends to produce a more accurate model instead of directly using output predictions. The student model is a regular model similar to the Π Model, and the teacher model has the same architecture as the student model with exponential moving averaging the student weights. Its two outstanding problems need to be urgently solved: (1) The reliability of the teacher model consistently surpassing that of the student model cannot be guaranteed. Inaccurate pseudo labels pose a substantial risk to the effective training of the student network. (2) The common reliance on consistency loss based on L1 distance introduces a challenge. The rigorous nature of the “strict” L1 loss function may incline the model towards overfitting inaccurate predictions, potentially resulting in confirmation bias.

### 2.4. Lightweight Neural Network

In recent years, there has been a significant surge in research efforts focused on developing lightweight neural network architectures to address the computational and memory constraints associated with deploying deep learning models in resource-constrained environments. Several innovative architectural designs have emerged, aiming to strike a balance between model efficiency and performance across various tasks. One prominent line of research involves the exploration of efficient building blocks that can be integrated into neural network architectures. The MobileNet architecture, proposed by Sandler et al. [32], introduced depthwise separable convolutions as a key component, significantly reducing the computational cost while maintaining competitive accuracy. Building upon this concept, subsequent works, such as MobileNetV2 [33] and ShuffleNet [34], further refined the use of lightweight structures like inverted residuals and channel shuffling to improve model efficiency. EfficientNet [35] represents another milestone in lightweight architecture research by introducing a compound scaling method. This method systematically scales up the model’s depth, width, and resolution, achieving state-of-the-art performance with significantly fewer parameters compared to traditional architectures. Beyond these foundational works, ongoing research explores novel neural network architectures, including EfficientDet [36], which extends the principles of EfficientNet to object detection tasks. This adaptation highlights the versatility of lightweight architectures in addressing diverse application domains, while emphasizing the importance of task-specific optimizations. Nevertheless, within the domain of highlight removal, there exists a paucity of relevant research.

## 3. Proposed Semi-TSGAN

### 3.1. The Framework of Semi-Supervised Highlight Removal

The proposed Semi-TSGAN amalgamates a semi-supervised learning paradigm based on the mean-teacher model and a generative adversarial network, implementing lightweight regulations. Within the Semi-TSGAN framework, semi-supervised learning endeavors to imbue the learning system with the capability to learn from both labeled and unlabeled datasets. To ensure the trustworthiness of pseudo labels assigned to unlabeled data, we establish a dependable repository dedicated to storing top-performing teacher outputs evaluated by NR-IQA. These dependable pseudo labels serve as a reliable compass for guiding the student’s training process through the incorporation of unsupervised teacher–student consistency loss $L_{un}^{\prime }$ and contrastive loss $L_{cr}$ mechanisms. The student’s weights are iteratively adjusted by minimizing both supervised and unsupervised losses. Furthermore, the teacher model undergoes updates via exponential moving average (EMA) derived from the student’s parameters. The schematic representation of this framework is elucidated in Figure 2. Formally, the highlight removal problem is cast as follows: The overarching objective is to learn a mapping on $D=D_{L}\cup D_{U}$ that transforms a given highlight image x into its clean counterpart y. We denote $D_{L}={\left\{(x_{i}^{l},y_{i}^{l})|x_{i}^{l}\in I_{h}^{l},y_{i}^{l}\in I_{f}^{l}\right\}}_{i=1}^{N}$ as the labeled dataset, where $x_{i}^{l}$ and $y_{i}^{l}$ represent the highlight image and the corresponding clean ground truth sourced from the degraded set $I_{h}^{l}$ and ground truth set $I_{f}^{l}$, respectively. Analogously, let $D_{U}={\left\{x_{i}^{u}|x_{i}^{u}\in I_{h}^{u}\right\}}_{i=1}^{M}$ denote the unlabeled dataset, featuring $x_{i}^{u}$ as the highlight image sampled from the degraded set $I_{h}^{u}$. It is imperative to note that the datasets represented by $D_{L}$ and $D_{U}$ are mutually exclusive (i.e., $D_{L}\cap D_{U}=\emptyset$).

Our semi-supervised learning framework adheres to the conventional configuration established in semi-supervised learning methodologies, as delineated in Figure 2. Specifically, our Semi-TSGAN comprises two networks sharing an identical structure, denoted as the teacher model and the student model respectively. The primary distinction between these two networks lies in their respective weight update mechanisms. The weights of the teacher $\theta_{t}$ undergo updating via exponential moving average (EMA) applied to the student’s weights $\theta_{s}$ with the momentum η∈(0,1)
(1)$$\theta_{t}=\eta \theta_{t}+(1-\eta )\theta_{s}$$

Employing this update strategy enables the teacher model to aggregate promptly previously acquired weights following each training iteration. As underscored in prior research, the adoption of temporal weight averaging serves to stabilize the training process, thereby potentially enhancing performance relative to standard gradient descent. In contrast, the weights of the student network $\theta_{s}$ are updated utilizing the gradient descent approach. Here we denote $L_{\sup}=\sum_{i=0}^{N}\left|f_{\theta_{s}}(x_{i}^{l})-y_{i}^{l}\right|$ as the supervised loss, while $L_{un}=\sum_{i=0}^{M}\left|f_{\theta_{s}}(\phi_{s}(x_{i}^{u}))-f_{\theta_{t}}(\phi_{t}(x_{i}^{u}))\right|$ represents the unsupervised teacher-student consistency loss, with $\left|\cdot \right|$ referring to the L1 distance. Additionally, $\phi_{s}$ and $\phi_{t}$, respectively, denote the data augmentations applied to the input of the student and teacher model. Typically, the optimization of the student network is formulated as the minimization of the following weighted addition of $L_{\sup}$ and $L_{un}$:(2)$$L_{\mathrm{total}}=L_{\sup}+\lambda L_{un}$$

In an ideal scenario, given that the teacher network generally exhibits superior performance compared to the student network, leveraging the teacher’s outputs ${\hat{y}}_{i}^{u}=f_{\theta_{t}}(\phi_{t}(x_{i}^{u}))$ as pseudo labels offers effective supervision for training the student network on the unlabeled dataset. However, it is imperative to acknowledge that the consistent superiority of the teacher’s outputs over those of the student is not guaranteed, and incorrect pseudo labels may potentially compromise the training of the student network.

### 3.2. NR-IQA Based Dependable Repository for Reliable Consistency

To tackle the aforementioned challenge, our strategy involves the meticulous selection of reliable teacher outputs to serve as pseudo labels. In domains like image classification and semantic segmentation, evaluating the reliability of network outputs typically involves metrics such as entropy and confidence. However, the extrapolation of these metrics to image restoration challenges proves nontrivial due to the unique characteristics inherent in such tasks. Notably, highlight removal, functioning as a regression task, demands to constrain the extent of luminance within the highlighted region to ensure the preservation of optimal brightness levels, thereby facilitating the retention of intricate details and textural nuances. To address this, we introduce a dependable repository designed to archive the optimal teacher network outputs throughout the training process. Concretely, we initialize our dependable repository (denoted as ℚu=∅) as an empty set. In each training iteration, we conduct a comparative analysis between the current output of the teacher, the student’s output, and the pseudo label residing in the dependable repository. If the teacher’s output attains superior quality, we substitute the pseudo label in the dependable repository with the teacher’s current output. This iterative process allows us to maintain a dependable repository (denoted as ℚu={yib}i=1M), wherein D′=DU∪ℚu={(xiu,yib)}i=1M represents the pseudo-labeled dataset. This dependable repository effectively records the most accurate pseudo labels, mitigating the inclusion of erroneous labels in the computation of the unsupervised consistency loss (denoted as Lun). Consequently, we can reformulate Lun in Equation (2) as follows:(3)Lun′=∑i=0Mfθs(ϕs(xiu))−yib

This gives rise to the pivotal query: How does one ascertain the quality of a prediction in the absence of an authentic label? Instinctively, one may turn to non-reference image quality assessment (NR-IQA) as an avenue for evaluation. Regrettably, the prevalent metrics prove insufficient in faithfully gauging the quality of highlight-free image restoration. Consequently, the establishment of a dependable metric repository grounded in such measures becomes a matter of uncertainty. In pursuit of the optimal NR-IQA framework for appraising restored highlight-free images, we undertake an empirical scrutiny of various NR-IQA metrics.

Given a degrade highlight image $x^{l}$ and a paired clean image $y^{l}$, we perform various linear combinations of them to get a set of images with different quality. Specifically, let $\alpha_{i}=0.1\times i,i=1,2,\ldots ,10$, we can obtain a set of ten images ${\left\{\alpha_{i}x^{l}+(1-\alpha_{i})y^{l}\right\}}_{i=1}^{10}$. With the increase of $\alpha_{i}$, the visual quality of the corresponding image deteriorates, as shown in Figure 3. It thus makes sense to evaluate the NR-IQA metrics based on how well they capture this monotonicity law. In particular, an NR-IQA metric is identified as reliable if its score on the $\alpha_{i}x^{l}+(1-\alpha_{i})y^{l}$ decreases with the increase of $\alpha_{i}$.

Adhering to this principle, we perform experiments utilizing several No-Reference Image Quality Assessment (NR-IQA) methodologies on the SHIQ dataset encompassing a diverse array of highlight scenes. The resultant experimental data are depicted in Figure 4. It becomes evident that MUSIQ [16], a deep learning-based approach, most closely aligns with the monotonicity law. Consequently, it is chosen as the metric to gauge the reliability of the networks’ outputs.

### 3.3. Contrastive Regularization for Confirmation Bias Reduction

Typically, many mean teacher-based methods employ the L1 distance as the consistency loss, as $L_{un}$ illustrated in Equation (2). However, this simplistic consistency loss has the potential to induce overfitting of the student model to erroneous predictions, leading to confirmation bias. In response to this challenge, we introduce contrastive loss during training. Contrastive learning, recognized as an effective paradigm in the self-supervised domain, aims to encourage a model to generate similar representations for positive pairs and dissimilar representations for negative pairs. Although it has witnessed significant success, its application to image restoration problems has become an area of recent exploration.

Despite its achievements, existing methodologies often construct contrastive loss on paired datasets, where positive and negative samples correspond to labels and degraded images, respectively. In this section, we propose the incorporation of contrastive loss in handling unlabeled data. To accomplish this, the construction of positive and negative pairs becomes crucial. While the paper suggests the direct use of the teacher’s output as positive samples [37], the issue of incorrect labeling, as discussed in previous sections, cautions against employing the teacher’s outputs in this manner. Indeed, the model plays a pivotal role in enhancing the robustness against noise samples. Notably, Semi-TSGAN demonstrates a robust design. A crucial aspect lies in the utilization of this contrastive loss, a loss function aimed at enhancing robustness by augmenting the Euclidean distance between dissimilar samples while diminishing the distance between similar ones. Such an approach finds common application in visual authentication domains such as pedestrian recognition.

Thanks to our introduced Dependable Repository, housing samples of potentially superior quality compared to the student’s outputs, we designate $y_{i}^{b}$ as our positive sample. Concurrently, adhering to the approach outlined in [37], we select the strongly augmented degraded image $\phi_{s}(x_{i}^{u})$ as our negative sample. Subsequently, with the positive and negative samples in place, the contrastive loss is computed as follows:(4)$$L_{cr}=\sum_{j=1}^{K}\sum_{i=1}^{M}\omega_{j}\frac{\left|\varphi_{j}({\tilde{y}}_{i}^{u}),\varphi_{j}(y_{i}^{b})\right|}{\left|\varphi_{j}({\tilde{y}}_{i}^{u}),\varphi_{j}(\phi_{s}(x_{i}^{u}))\right|}$$
where ${\tilde{y}}_{i}^{u}=f_{\theta_{s}}(\phi_{s}(x_{i}^{u}))$ is the student’s prediction on the unlabeled dataset $D_{U}$. $\varphi_{j}(\cdot )$ represents the $j_{th}$ hidden layer and $\omega_{j}$ is the weight coefficient. We use L1 loss to measure the distance in feature space between the students’ outputs with the positive $y_{i}^{b}$ samples and negative samples $x_{i}^{u}$.

### 3.4. Lightweight Model Description

In this part, we design Lightweight module for highlight removal in generative network of Semi-TSGAN. In delineating the mechanism of highlight image degradation, it can be seen that it is closely related to mirror reflection and diffuse reflection for highlight images. Moreover, it exhibits correlations with multiple variables such as materials, light intensity, and angle. Consequently, various specific subnetworks are required to meet the task if taking these variables into consideration, thereby inevitably augmenting the overall complexity and parameters of the model, and it is not easy to train and for deployment. Therefore, in this paper, our highlight removal model can be simplified as follows:(5)$$J\left(x\right)=I\left(x\right)-H\left(x\right)$$
where $J\left(x\right)$ denotes highlight-free image, $I\left(x\right)$ denotes highlight image, $H\left(x\right)$ denotes highlight degradation noise. This paper generates the design of the network generator model as the end-to-end form, allowing it to directly learn the residual mapping relationship between highlight-free image $J\left(x\right)$ and highlight image $I\left(x\right)$. The overall structure of the network is shown in Figure 5. The discriminator structure is consistent with the paper [38], and this part will not be repeated. The presented section illustrates the generator part of a generative adversarial network, featuring a direct connectivity structure comprising two 3 × 3 convolutional layers, along with a suite of feature extraction modules and attention reinforcement modules. Notably, in numerous generator models, diverse upsampling and subsampling operations are typically employed for designing distinct modules, enabling the model to acquire more extensive semantic information. However, this approach frequently results in an imbalanced channel width and incongruent feature maps, hindering the optimization of a lightweight network size and impeding improvements in inference speed. Consequently, for both the characteristic extraction module and the attention reinforcement module, upsampling and subsampling operations are deliberately omitted, ensuring the consistency of feature map characteristics between the input and output. The subsequent section, dedicated to network structure design, introduces the full-scale feature aggregation module within the characteristic extraction module, along with the comprehensive attention module constituting an attention reinforcement module. The discussion encompasses their design principles and specific operations.

The full-scale feature aggregation module is one of the core components of the Semi-TSGAN generator model. The design of the overall structure follows the principles of the lightweight module. It is equipped with a common residual connection to reduce model complexity. As shown in Figure 6, for inputs with size H×W×C, we first use a point-by-point convolution to adjust the feature dimension. Then we use in-depth convolution with different sizes to extract more features of different scale and learn more comprehensive semantic information. In this module, the smallest in-depth convolution is 3 × 3, and the maximum in-depth convolution used is the 9 × 9 kernel. The reason for no longer using larger convolution kernels (such as 11 × 11) is that in the image recovery task, the larger kernel is limited to the model’s performance, and it will also lead to a significant increase in model complexity. This is not in line with the original intention of designing the Semi-TSGAN model generator model. Therefore, the largest convolution kernel size in the module is fixed at 9 × 9.

When the convolutional size of the module is determined, two new mapping operations are further designed, which are named the cascade mapping and parallel mapping, respectively, as shown in the green and orange arrows in Figure 6. Cascade mapping and parallel mapping can fully interact and integrate the extracted features within the module. The cascade mapping connects the features of different in-depth convolution extraction in turn, and the connection order follows the principle of convolution kernel size from small to large. There are two aspects of the benefits of this design. First, it can alleviate the structural redundant problem that is common in multi-branch modules to a certain extent. Generally speaking, the full stack of ordinary multi-branch modules will cause the phenomenon of “similar scale redundancy” in the generated feature diagram. This is not conducive to the optimization of parameters in the Semi-TSGAN model generator model. The cascade mapping operation in this paper can provide more flexibility for the information flow between different branches and enable them to realize the full information interaction process. Therefore, more multi-scale information will be retained in the feature map. Second, the cascade mapping has expanded the module’s receptive field. This feature is more conducive to the Semi-TSGAN to learn global information with highlight images. Specifically, for ordinary multi-branch modules, the most common approach is to use directly the 9 × 9 large kernel or stack four 3 × 3 small convolution kernels. But no matter which way, the maximum receptive field inside of the module is only 9 × 9. However, for the full-scale feature aggregation module in this paper, the receptive field can be represented by Equation (6):(6)$$R=\sum_{k=1}^{n}l_{k}\times l_{k},s.t.l_{k}=l_{k-1}+((f_{k}-1)\times \prod_{i=1}^{k-1}s_{i})$$
where $l_{k}$ represents the receptive field of the k layer, $f_{k}$ represents the kernel size, and $s_{i}$ means stride size. In the full-scale feature aggregation module, if $f_{1}=3$, $f_{2}=5$, $f_{3}=7$, $f_{4}=9$, $s_{i}=1$, the receptive field of the full-scale feature aggregation module is 21 × 21. Compared with ordinary multi-branch modules, the receptive field of the full-scale feature aggregation module has expanded by 5.4 times.

Parallel mapping operations can dynamically integrate the independent characteristics generated by different convolution kernels. As shown in the orange arrows in Figure 6, a new sharing and gathering gate module is embedded in the parallel mapping operation. Sharing and gathering gate modules are linked to a small neural network that contains global average pooling layers, a multi-layer perceptron, and *sigmoid* activation functions. Its output is a set of vectors, not a separate value. This enables more fine-grained information to integrate into the characteristic channels of each group. In addition, for the characteristics of each group, sharing and gathering gate modules have always had the characteristics of parameter sharing, which is also conducive to reducing the total number of full-scale feature aggregation modules to a certain extent.

As a new lightweight module, the full-scale feature aggregation module has less learning parameters itself. Consequently, merely incorporating it as an intrinsic composition module within the Semi-TSGAN generator model is bound to exert a discernible influence on the overall model performance. In this case, the introduction of attention mechanism is very conducive to improving the performance of lightweight models, because the attention mechanism can help the model pay attention to more effective features or information. Following the idea of full-scale characteristics in the full-scale feature aggregation module, we further designed a new full-scale attention module. The overall structure of the module is shown in Figure 7.

It can be seen that the overall structure of the full-scale attention module is based on a residual structure in the dual form. This method of dual residual connection has two advantages. First, it is conducive to increasing the circulation of potential information flow between full-scale feature aggregation modules and full-scale attention modules. Second, it is conducive to the follow-up full-scale attention mechanism to learn more important image features in different dimensions. In the later part, we introduce the three parts of the core attentive mechanism in the full-scale attention module in turn, which are channel attention, pixel attention, and space attention.

Figure 8 shows the channel attention mechanism in the full-scale attention module. The role of the channel attention mechanism is to allocate different weights for the characteristics of different channels because the deep features from the feature extraction module often have different importance in the channel dimension. It can be seen that the channel attention mechanism first uses the global average pooling layer to generate the global information of the channel, and then the convolution and the activation function generate the intermediate feature diagram and finally use the activation function to obtain weight. The calculation process of channel attention is shown Equation (7):(7)$$\begin{array}{l} \mathrm{CA}(x)=\delta (\mathrm{PConv}(\mathrm{NA}(\mathrm{PConv}(\mathrm{Avg}(x)))))\\ F_{\mathrm{CA}}=x⊗\mathrm{CA}(x) \end{array}$$
where, x denotes the input of feature map, Avg denotes average pooling, PConv denotes point-wise convolution, NA denotes LeakyReLU nonlinear activation function, δ represents Sigmoid function, $F_{\mathrm{CA}}$ represents the output of channel attention, ⊗ represents element-wise multiplication.

In addition to channel attention, pixel attention is another part of the full-scale attention module. Generally speaking, the background of highlight images is complicated, so the important features of small and small goals in the background image are often difficult to extract. At this time, the introduction of pixel attention will help the model to learn the fine particle characteristics or information in high-gloss images. The pixel attention mechanism in the full-scale attention module is shown in Figure 9. It can be seen that pixel attention directly uses the output of channel attention as its own input, and then uses two points of convolution and activation functions and activating functions to generate weight. The calculation process of pixel attention is shown in Equation (8):(8)$$\begin{array}{l} \mathrm{PA}(F_{\mathrm{CA}})=\delta (\mathrm{PConv}(\mathrm{NA}(\mathrm{PConv}(F_{\mathrm{CA}}))))\\ F_{\mathrm{PA}}=F_{\mathrm{CA}}⊗\mathrm{PA}(F_{\mathrm{CA}}) \end{array}$$
where, $F_{\mathrm{CA}}$ denotes the input of pixel attention, PConv denotes point-wise convolution, NA denotes LeakyReLU nonlinear activation function, δ represents Sigmoid function, $F_{\mathrm{PA}}$ represents the output of pixel attention, ⊗ represents element-wise multiplication.

Finally, for highlight images, the distribution of highlight is often uneven. The presence of high-frequency parts or highlight parts in the image increases the difficulty of the model to highlight removal. Therefore, the space attention mechanism is designed to consider the different importance of different spaces in the image as shown in Figure 10.

First, the output of the pixel attention mechanism is used as the input of the space attention mechanism. Then we perform average pooling and maximum pooling operations, and concatenate the pooling results on the channel dimension. Finally, we use points and functions to generate attention diagrams. The calculation process of spatial attention is shown in Equation (9):(9)$$\begin{array}{l} \mathrm{SA}(F_{\mathrm{PA}})=\delta (\mathrm{PConv}(\mathrm{Concat}[(\mathrm{Avg}(F_{\mathrm{PA}});\mathrm{Max}(F_{\mathrm{PA}}))]))\\ F_{\mathrm{SA}}=F_{\mathrm{PA}}⊗\mathrm{SA}(F_{\mathrm{PA}}) \end{array}$$
where, $F_{\mathrm{PA}}$ denotes the input of spatial attention, Avg and Max denote average pooling and max pooling, respectively, Concat denotes concatenation operation, δ represents Sigmoid function, $F_{\mathrm{SA}}$ represents the output of spatial attention, ⊗ represents element-wise multiplication.

### 3.5. Overall Optimization Objective

Similar to Equation (2), our final optimization objective consists of supervised loss and unsupervised loss. For the supervised loss, unlike the one defined in Equation (2) that only calculates the L1 distance, we follow [39] to extend the original $L_{\sup}$ by adding perceptual loss $L_{\mathrm{per}}$ and gradient penalty $L_{\mathrm{grad}}$:(10)$$L_{\sup}^{\prime }=L_{\sup}+\beta_{1}L_{\mathrm{per}}+\beta_{2}L_{\mathrm{grad}}$$

For the unsupervised loss, we replace the original $L_{un}$ by a combination of the proposed reliable teacher–student consistency loss and contrastive loss:(11)$$L_{un}^{\prime \prime }=L_{un}^{\prime }+rL_{cr}$$

Finally, we rewrite our overall optimization objective following Equation (2):(12)$$L_{\mathrm{overall}}=L_{\sup}^{\prime }+\lambda L_{un}^{\prime \prime }$$

## 4. Experimental Results

### 4.1. Implementation Details

Our approach is implemented using the PyTorch library [40] and executed on NVIDIA RTX 4090 GPUs (NVIDIA, Santa Clara, CA, USA). The AdamP optimizer [41] is employed for its rapid convergence to the optimum, chosen primarily to expedite the training process. Throughout training, a minibatch size of 16 is utilized, with eight samples designated as labeled dataset and eight as unlabeled. The initial learning rate is configured at 2 × 10^−4^. The training regimen spans 200 epochs, with the learning rate subjected to a 0.1 multiplication at the 100th epoch. All training images undergo cropping to a standardized size. In the context of data augmentation for unlabeled data, resizing is exclusively applied to the teacher’s inputs, while robust data augmentation techniques, including resize, color jitter, Gaussian blur, and grayscale transformations, are imposed on the student’s inputs. Labeled data undergo standard augmentation, encompassing resize, random crop, and rotation. The weights assigned to different loss components are determined as follows. The parameters denoted as $\beta_{1}=0.3,\beta_{2}=0.1,r=1$ and λ are updated with the training epoch t, following an exponential warming-up function [42] $\lambda (t)=0.2\times e^{-5{\left(1-t/200\right)}^{2}}$.

### 4.2. Datasets and Metrics

SHIQ: The SHIQ dataset is meticulously curated for the explicit purpose of facilitating highlight detection and removal, featuring meticulous ground truth annotations. Captured in diverse natural scenes, SHIQ encapsulates a rich array of illumination conditions, object materials, and scenes, rendering it a comprehensive and multifaceted resource for our research endeavors. This dataset comprises approximately 12,000 image triples, with each triple encompassing a highlight image, a corresponding ground truth, and an accompanying highlight mask, all formatted at a resolution of 200 × 200. In our experiments, 6000 pairs of labeled images and 3000 unlabeled images were selected for model training, and 1000 labeling data sets were selected for them for testing. The test set is meticulously constructed, incorporating both full-reference and non-reference benchmarks. The full-reference test set includes 1000 pairs denoted as testF. Meanwhile, the non-reference test set comprises nearly 1000 real-world highlight images without accompanying ground truths denoted as testNF.

RD: The RD dataset meticulously gathers a corpus comprising 2025 meticulously paired images. Each pair encompasses an image featuring text-aware specular highlights, its corresponding counterpart devoid of highlights, and a binary mask image meticulously delineating the precise locations of these highlights. The imagery encapsulates various document types such as ID cards and driver licenses, characterized by substantial textual information. This deliberate inclusion of text-rich content ensures the dataset’s relevance to scenarios where text-aware specular highlights pose significant challenges. In our experimental protocol, we meticulously curated a dataset comprising 1200 pairs of labeled images for supervised training, supplemented by an additional 600 unlabeled images. Concurrently, we reserved 400 labeled datasets for model testing. The test set, designed for comprehensive evaluation, encompasses both full-reference and non-reference benchmarks. The full-reference test set, denoted as testF, comprises 200 pairs of images with corresponding ground truths. Simultaneously, the non-reference test set, denoted as testNF, consists of nearly 200 real-world highlight images without associated ground truth information. This robust dataset configuration ensures a thorough assessment of the model’s performance under diverse scenarios.

PSD: The PSD dataset comprises a comprehensive collection of 13,380 images, each captured across 2210 distinct scenes. Diverse objects and backgrounds are employed to construct this dataset. It is categorized based on two distinct polarization conditions. The first category encompasses 1010 sets of images captured at fixed polarization angles, while the second category includes 1200 image pairs captured with random polarization angles. In the former, each group is constituted of 12 images, each photographed at a specific fixed polarization angle. In the experimental phase, we curated a dataset consisting of 800 pairs of labeled images and 454 unlabeled images for model training. Additionally, 54 labeled datasets were specifically set aside for model testing. The test set was meticulously designed to encompass both full-reference and non-reference benchmarks. The full-reference test set, designated as testF, comprises 200 image pairs with corresponding ground truths. Simultaneously, the non-reference test set, denoted as testNF, comprises almost 200 real-world highlight images without associated ground truth information. This dataset configuration adheres to rigorous standards to ensure a comprehensive evaluation of the model’s performance across various scenarios.

Metrics: In alignment with prevalent methodologies observed in contemporary research endeavors [43], we adopt the widely recognized Peak Signal-to-Noise Ratio (PSNR) and Structural Similarity Index (SSIM) as the cornerstone metrics for performance assessment in our experiments. Larger values of both metrics denote superior performance, adhering to the established conventions in the field.

Extra statement: The quality of the samples is crucial, particularly within the context of Semi-supervised learning. The ratio of labeled to unlabeled samples varies depending on the dataset, with some datasets exhibiting intrinsic characteristics while others are artificially constructed. Nevertheless, regardless of the ratio between labeled and unlabeled samples, appropriate adjustment of λ can be achieved.

### 4.3. Comparison with the State-of-the-Arts

In our evaluative framework, we conduct a comprehensive comparison between our proposed Semi-TSGN and several state-of-the-art highlight removal methods, namely Tan [44], Yang [45], Shen [24], Akashi [23], Shi [27], Yi [46], Huang [47], and Fu [5] using the SHIQ/RD/PSD dataset. To the best of our knowledge, we first propose a Semi-supervised learning paradigm for highlight removal. Conversely, Shi [27] and Fu [5] adopt a supervised paradigm, leveraging paired datasets that encompass highlight and highlight-free images, in conjunction with corresponding masks and specular reflections for training purposes. All compared methods undergo retraining on our dedicated training dataset. For the full-reference test sets, we perform quantitative assessments utilizing PSNR and SSIM [43].

The tabulated data in Table 1 illustrate the quantitative outcomes across SHIQ, RD, and PSD datasets. Specifically, Table 1 provides a comprehensive overview of the quantitative results concerning testF. The superior performance of our method is noteworthy, particularly in terms of PSNR and SSIM metrics. This performance divergence might be attributed to the inclusion of unlabeled real highlight images during the training process, potentially influencing the network to give precedence to real highlight scenes. This hypothesis gains further credence from the quantitative results observed in testF, where our method demonstrates a significant 6 dB advantage in PSNR over the second-best method. In addition to the quantitative findings, Figure 11a–c displays the qualitative outcomes on the SHIQ/PSD/RD dataset. Our results exhibit visually pleasing attributes, characterized by enhanced texture and intricate details, in contrast to the comparative methods which exhibit deficiencies in these aspects.

Meanwhile, the non-reference test set undergoes evaluation using DISTS, NR-IQA_DMetaLearning, MB-CNN, and MUSIQ. Quantitative results on MUSIQ, DISTS, NR-IQA_DMetaLearning, and MB-CNN are outlined in Table 2. A cursory examination of the table reveals the substantial outperformance of our method, particularly evident in MUSIQ. Competitive performance is also achieved in MB-CNN. However, it is important to note that MB-CNN may exhibit bias towards specific characteristics, potentially limiting their ability to accurately reflect the genuine visual quality of restored images. Similarly, the interpretability of MUSIQ serves as a reference, indicating that quantitative results alone may not fully capture the quality of restored highlight images due to the nascent nature of NR-IQA metrics. To further underscore our approach’s efficacy, qualitative results on the four non-reference metrics are presented in Figure 12. Comparative analysis reveals that our framework robustly restores various types of highlight images, exhibiting natural textures and intricate details. Guided by music demonstrates notable generalization across diverse highlight scenes.

### 4.4. Ablation Study

#### 4.4.1. Semi-Supervised Learning Framework Study

To assess the efficacy of Semi-TSGAN, we conducted ablation studies to elucidate the impact of key components in our methodology. The distinct configurations are delineated as follows: (a) Sup-base: wherein the GAN network is trained without incorporating Semi-supervised learning and unlabeled data. (b) Semi-base: representing the base Semi-supervised training with the inclusion of the consistency loss Lun. (c) Semi-base+DR*: involving a dependable repository based on Semi-base without the application of contrastive loss. (d) Semi-base+CL*: introducing contrastive loss to Semi-base, but excluding the utilization of a dependable repository. (e) Semi-TSGAN: denoting our proposed Semi-TSGAN. The quantitative outcomes of these distinct configurations on SHIQ dataset are summarized in Table 3, revealing the superior performance of our comprehensive solution. Furthermore, comparisons between Semi-base+DR* and Semi-base, as well as Semi-TSGAN and Semi-base+CL*, substantiate the efficacy of integrating the dependable repository. The qualitative results on SHIQ/PSD/RD dataset, presented in Figure 13a–c, warrant particular attention, with a focus on Semi-base+CL* and Semi-base+DR*. Notably, in Semi-base+CL*, the absence of reliable positive samples exacerbates the impact of the contrastive loss, leading to pronounced deviations from the negative samples (inputs) and consequential over-restoration. Conversely, in Semi-base+DR*, the deficiency of contrastive loss manifests as detail distortions, and the restored images remain proximate to the degraded inputs. These two ablation studies collectively underscore the utility of both the dependable repository and contrastive regularization in enhancing the overall performance of Semi-TSGAN.

#### 4.4.2. Lightweight Model Study

(1) Kernel configurations analysis of the full-scale feature aggregation in the SEMI-TSGAN generator.

In the realm of lightweight modules, the configuration of the convolution kernel stands out as a key area of research focus. Consequently, this section undertakes a preliminary analysis of the effects stemming from the convolution kernel configuration within the full-scale feature aggregation module (simply denoted as FSFA) on model performance of the SHIQ dataset (The analysis data later give also the result on this data set). The ensuing experimental results are presented in Table 4.

Table 4 provides a comprehensive overview of the parameter count and performance metrics for various convolutional layers employed in constructing the FSFA {orderConv} module. It is evident that the standard convolutional layer yields the highest total parameter count, reaching 2.15 × 10^5^. However, despite this substantial parameter count, its performance is not optimal. This observation underscores the inherent redundancy in standard convolution operations, rendering them unsuitable for directly constructing lightweight modules. In contrast, utilizing depth-wise separable convolution to build an FSFA {3_3_3_3} module yields a performance equivalent to that of the FSFA {orderConv} module using the standard convolutional layer. This indicates the effectiveness of employing depth-wise separable convolution in constructing lightweight modules. Further investigation reveals that the performance of multi-scale depth separable convolution surpasses that of a single-scale depth separable convolution. For instance, in the case of the FSFA {1_3_5_7} module, there is a nearly 3db improvement in the PSNR indicator compared to the FSFA {3_3_3_3} module, with a parameter count of only 1.23 × 10^4^. This underscores the significance of incorporating multi-scale convolution kernels to enhance the performance of lightweight modules. Subsequent experiments demonstrate that combining multi-scale depths of 3, 5, 7, and 9 achieves optimal results for the model. The rationale behind this lies in the superior ability of these convolutional kernels to extract effective features from fog images. Notably, convolutional kernels with excessive sizes, such as 11 × 11, hinder the detailed extraction of image features. Conversely, smaller convolutional kernels, such as 1 × 1, struggle to comprehend the global semantic information of the image. The combination of 3, 5, 7, and 9 emerges as a balanced choice in terms of size, enabling the model to effectively adapt to lightweight image highlight removal while preserving crucial details.

The aforementioned experimental results comprehensively illustrate and analyze the influence of diverse convolution kernel configurations on the efficacy of the full-scale feature aggregation module. In the construction of a lightweight module, it becomes evident that careful consideration of the convolution kernel size plays a pivotal role. This consideration significantly contributes to achieving a balance between model compactness and optimal performance, as reflected in the final parameter count and overall model performance within the neural network architecture. Thus, emphasizing the judicious selection of convolution kernel sizes proves instrumental in attaining the desired lightweight characteristics, while ensuring an optimal trade-off between model complexity and performance.

(2) Analysis of feature fusion strategies.

Fundamentally, the cascade mapping and parallel mapping techniques introduced in the FSFA module represent distinctive feature fusion strategies. To ascertain their effectiveness, we conducted comparisons with two widely employed fusion strategies: point-wise addition and matrix concatenation. The results of these experiments are presented in Table 5.

Table 5 illustrates a noteworthy enhancement in performance when employing the cascade and parallel mapping feature fusion strategy in comparison to two other widely used strategies: point-wise addition and matrix concatenation. Specifically, in contrast to point-wise addition, the cascade & parallel mapping strategy demonstrates a notable 3 dB improvement in PSNR. This improvement can be attributed to the limitations of point-wise addition, which operates solely at the element level and fails to fully consider the intricate interactions among the generated multi-scale features. Similarly, when compared to the matrix concatenation feature fusion strategy, cascade & parallel mapping exhibits a 2 dB increase in PSNR. This improvement arises from the inherent limitations of matrix concatenation, which merely concatenates feature maps along the channel dimension without establishing an effective connection between channels. Consequently, its effectiveness is sub-optimal. Furthermore, combining the characteristic fusion strategies of cascade & parallel mapping surpasses the individual use of cascade mapping or parallel mapping. The PSNR value sees a minimum increase of 2.6 dB, and the SSIM value rises from 0.97 to 0.98. This observation underscores that cascade & parallel mapping achieves a comprehensive integration of internal information within the FSFA module. This robust characteristic interaction ensures the superior performance of the FSFA module, showcasing the efficacy of cascade & parallel mapping in promoting meaningful feature interactions.

(3) Comparison with mature lightweight modules.

Given that the FSFA module operates as a plug-and-play lightweight module, this section’s experiment aims to compare its performance with other established lightweight modules, affirming its effectiveness in highlight image removal tasks. Three widely adopted lightweight modules were selected for comparison: Shuffle Unit [34], MixConv [48], and Xception Block [49]. Notably, these modules exhibit certain correlations with the proposed FSFA module. For instance, Shuffle Unit employs feature fusion methods akin to channel-washing techniques, MixConv leverages multi-scale concepts for module construction, and Xception Block incorporates a multi-branch structure in its design. Consequently, these three lightweight modules are characterized as feature extraction modules, and three Semi-TSGAN variants were formulated accordingly. The experimental outcomes are detailed in Table 6.

Table 6 clearly demonstrates the significant performance advantages of the proposed FSFA module compared to the three representative lightweight modules: Shuffle Unit [34], MixConv [48], and Xception Block [49]. The FSFA module showcases a minimum increase of 4.8 dB in PSNR, accompanied by an improvement in SSIM from 0.95 to 0.98. Notably, even when compared to the Xception Block, which exhibits the best performance among the selected lightweight modules, the FSFA module maintains distinct advantages in terms of the overall parameter count, amounting to only 67% of the Xception Block.

(4) Full-scale attention module.

The full-scale attention module is another important component module of this chapter lightweight highlight removal network. This section analyzes its effectiveness. First of all, decompose the various attention mechanisms in the full-scale attention module, and the specific description is as follows:

W/O: No attention mechanism.

CA, PA, SA: There is only one kind of attention mechanism, that is, channel attention, pixel attention or spatial attention.

CA & PA, CA & SA, PA & SA: The combination of two different attention mechanisms.

OSAtt: The full-scale attention module proposed.

The experimental findings, depicted in Figure 14, underscore the positive impact of incorporating attention mechanisms on model performance. Notably, the introduction of any category of attention mechanism leads to improvements, although the extent of contribution varies across different attention types. Spatial attention mechanisms, for instance, exhibit more substantial contributions to enhancing the PSNR indicator, resulting in improved image quality, while pixel attention mechanisms prove more beneficial for enhancing the SSIM indicator. However, the ultimate proposal of a full-scale attention module, which combines three distinct attention mechanisms, emerges as the most effective in improving various performance indicators. In comparison to the benchmark model devoid of any attention mechanism, the inclusion of the full-scale attention module results in a 0.87 dB increase in PSNR, and SSIM rises from 0.981 to 0.984. These experimental outcomes underscore the effectiveness of introducing attention mechanisms to enhance model performance in lightweight models. It is crucial, however, to emphasize the interplay of different levels of attention mechanisms, ensuring a harmonious integration that maximizes the overall performance of the lightweight model.

Furthermore, in our performance evaluation, the OSAtt module is introduced and compared with four prominent attention mechanism modules: SE attention module, CBAM attention module, SA attention module, and SK attention module. The comparative experimental results are presented in Table 7.

Table 7 reveals that among the four representative attention modules, the SK attention module demonstrates the highest experimental indicators, achieving a PSNR of 34.16 and an SSIM of 0.9790. The inference drawn suggests that the SK attention module employs convolution cores with distinct receptive fields, assigning different weights to parameters for adaptive handling. This unique flexibility and adaptability contribute to its commendable experimental results. However, the OSAtt module, proposed in this paper, surpasses the performance of the SK attention module, achieving a notable improvement. Specifically, the OSAtt module exhibits a 0.5 dB increase in PSNR and a rise in SSIM from 0.97 to 0.98. This improvement is attributed to the full-scale feature learning methodology within the OSAtt module, proving more adept at capturing image features of varying importance across different levels. Additionally, the OSAtt module reduces parameters by 16% compared to the SK attention module, further underscoring its efficacy and suitability for lightweight model construction. The feature aggregation module can significantly reduce the number of parameters, and the attention mechanism can improve the efficiency of parameter utilization, which increases the memory and computation loss to a certain extent, but this is a balance. Overall, the model is slimmer and performance has increased.

## 5. Conclusions

We introduced a highly-efficient and easily-deployment Semi-supervised method for highlight removal, denominated as Semi-TSGAN. The compelling performance demonstrated in ablation experiments underscores the method’s superiority over other state-of-the-art algorithms, attributable to the judicious integration of reliable teacher–student consistency and contrastive regularization learning framework and lightweight model design. Future Research Directions: The trajectory for subsequent research endeavors can be bifurcated into two pivotal directions: Expansion of Semi-Supervised Framework: extend the Semi-supervised framework to encompass diverse restoration tasks, broadening the applicability and versatility of the proposed methodology. Optimization of Memory Usage: delve into the optimization of memory usage during training, with a specific focus on enhancing performance through meticulous memory management strategies.

## Figures and Tables

**Figure 1 sensors-24-03090-f001:**
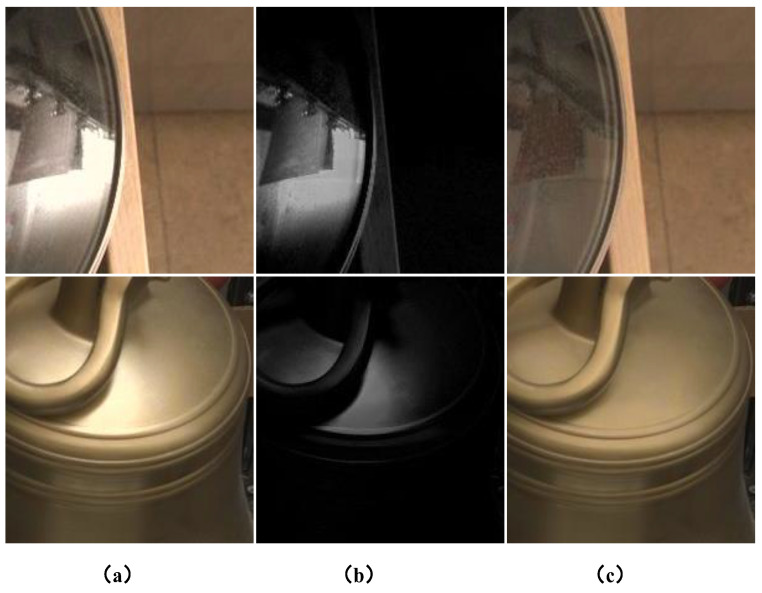
Examples from SHIQ. (**a**) Shows real-world highlight images, (**b**) shows highlight intensity images, and (**c**) shows real-world highlight-free images.

**Figure 2 sensors-24-03090-f002:**
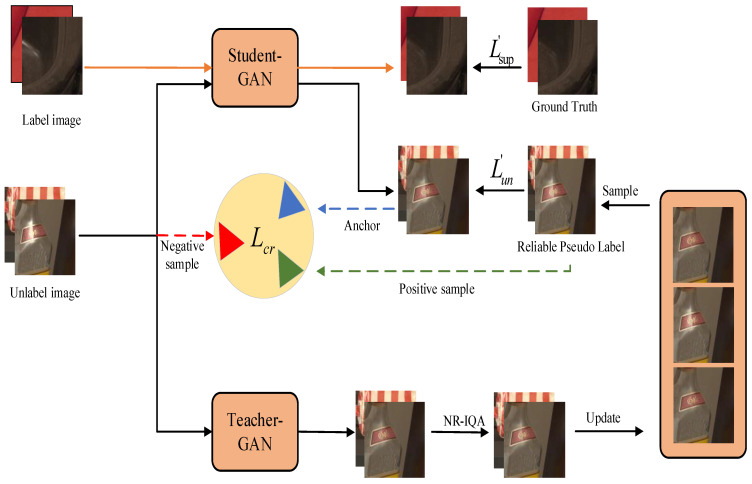
Illustration of our framework Semi-TSGAN.

**Figure 3 sensors-24-03090-f003:**
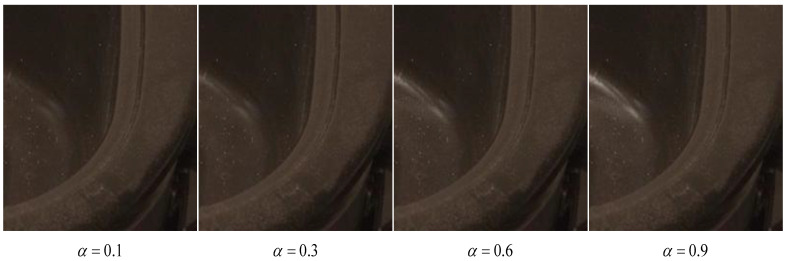
Examples of image fusion based on different α.

**Figure 4 sensors-24-03090-f004:**
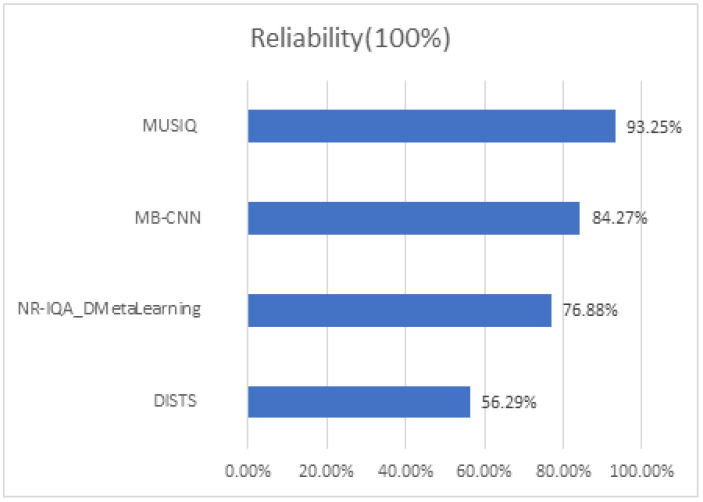
The results of different non-reference IQA indicators on SHIQ benchmark, including MUSIQ [16], DISTS [13], NR-IQA_DMetaLearning [14], MB-CNN [15].

**Figure 5 sensors-24-03090-f005:**
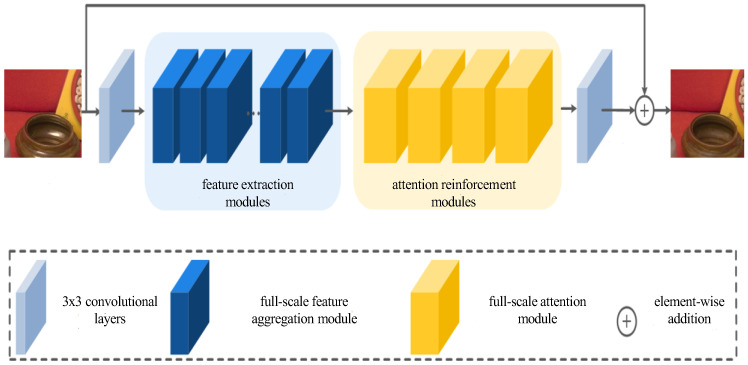
SEMI-TSGAN model generator structure.

**Figure 6 sensors-24-03090-f006:**
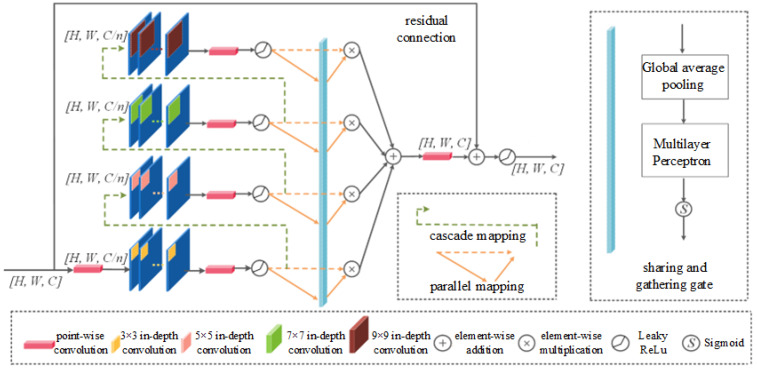
Full -scale feature aggregation module structure.

**Figure 7 sensors-24-03090-f007:**
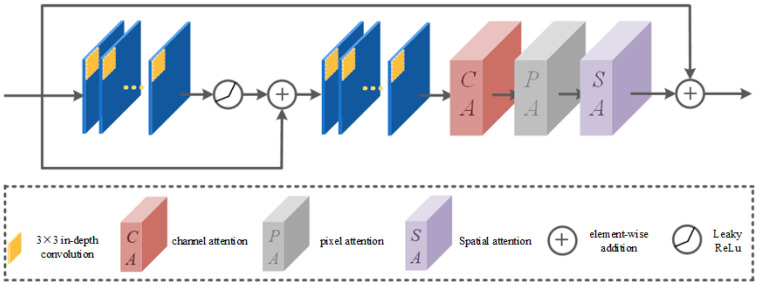
Full-scale attention module structure.

**Figure 8 sensors-24-03090-f008:**
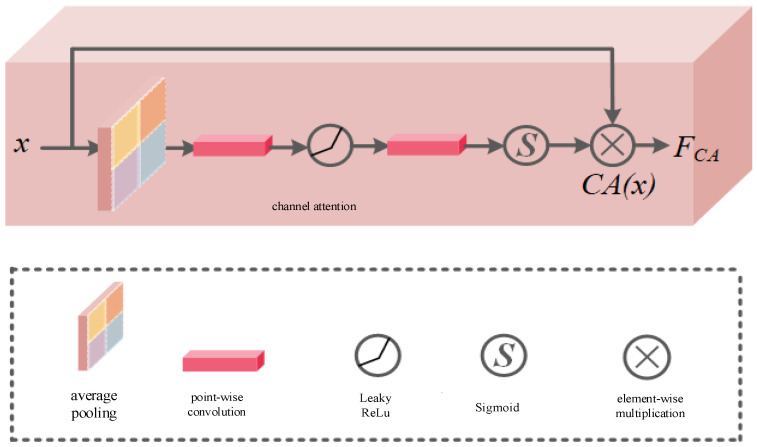
Channel attention module structure.

**Figure 9 sensors-24-03090-f009:**
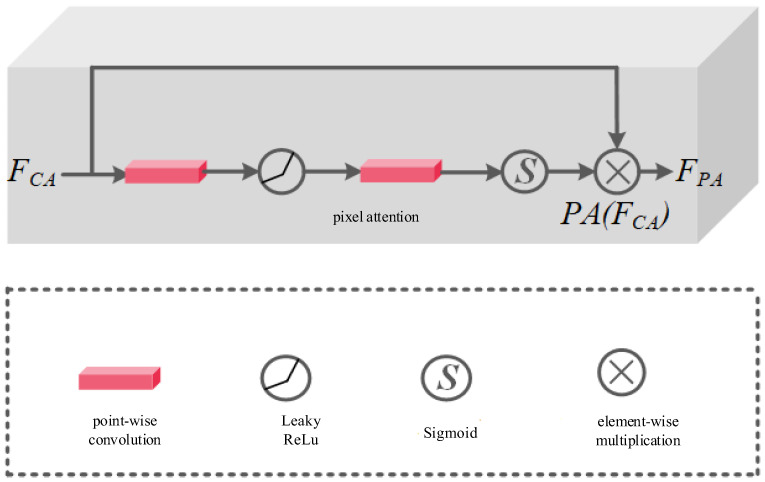
Pixel attention module structure.

**Figure 10 sensors-24-03090-f010:**
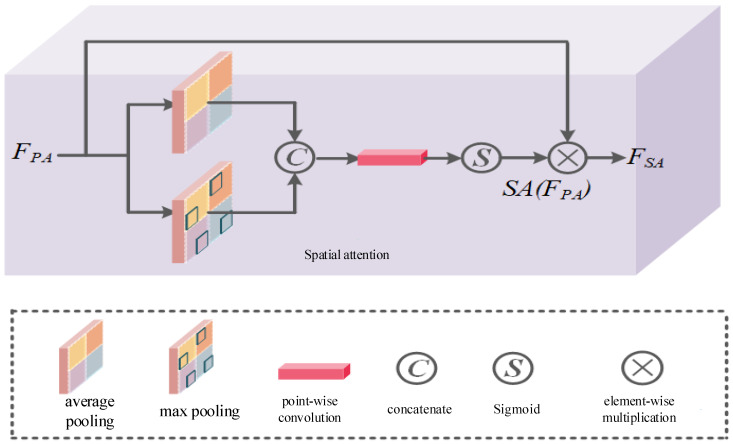
Spatial attention module structure.

**Figure 11 sensors-24-03090-f011:**
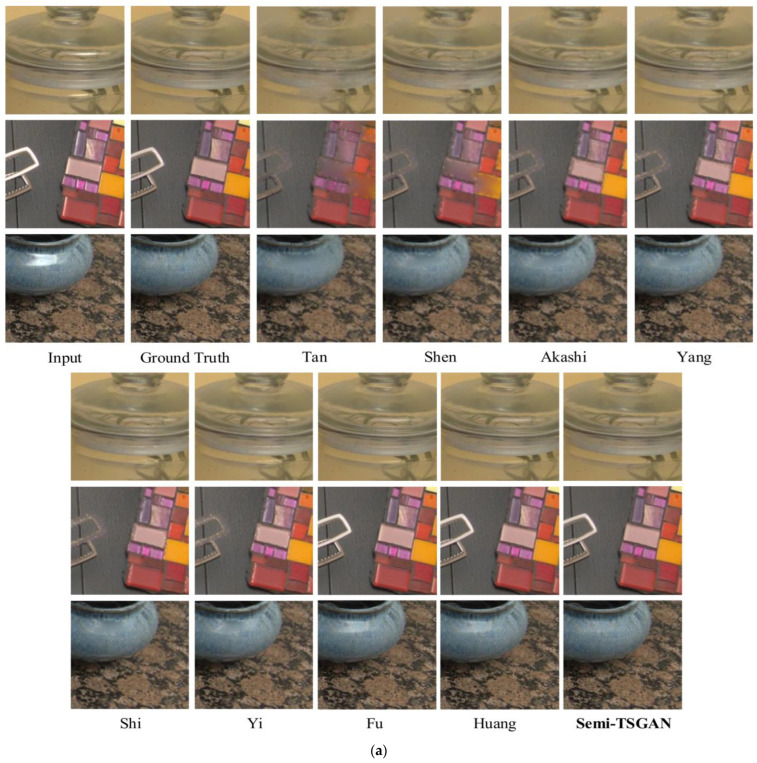
(**a**) Visual comparisons of full-reference data from SHIQ benchmark. (**b**) Visual comparisons of full-reference data from PSD benchmark. (**c**) Visual comparisons of full-reference data from RD benchmark. Fu et al. (2021) [5]; Akashi & Okatani (2014) [23]; Shen & Zheng (2013) [24]; Shi et al. (2017) [27]; Tan & Ikeuchi (2008) [44]; Yang et al. (2010) [45]; Yi et al. (2020) [46]; Huang et al. (2022) [47].

**Figure 12 sensors-24-03090-f012:**
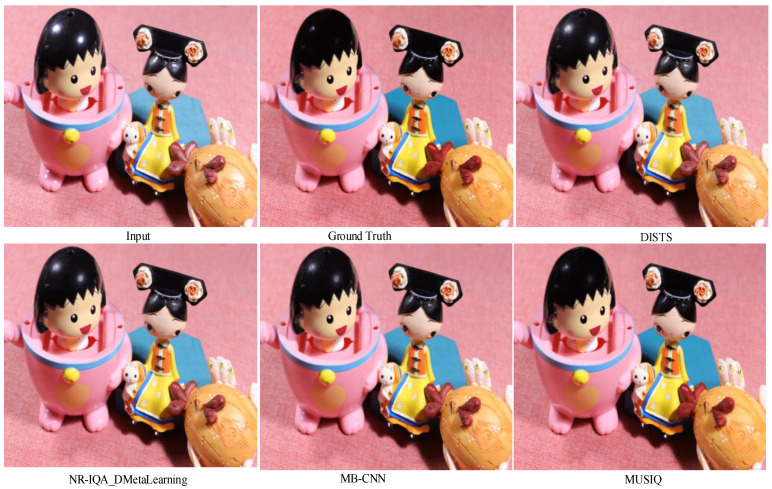
Visual comparisons of four non-reference metrics of Semi-TSGAN on PSD dataset.

**Figure 13 sensors-24-03090-f013:**
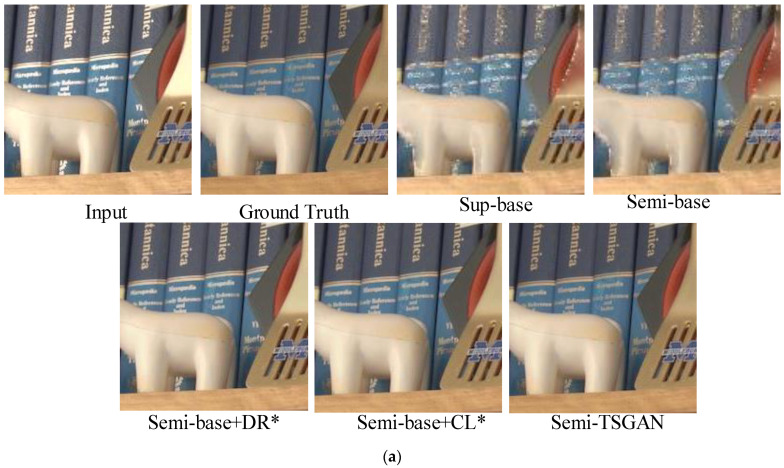
(**a**) Visual comparisons of diverse configurations on SHIQ benchmark. (**b**) Visual comparisons of diverse configurations on PSD benchmark. (**c**) Visual comparisons of diverse configurations on RD benchmark.

**Figure 14 sensors-24-03090-f014:**
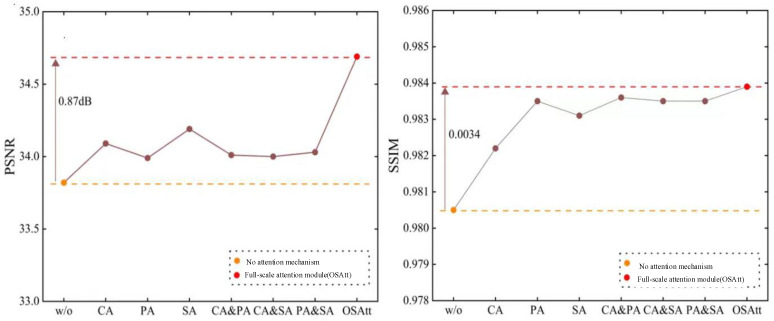
Performance metrics for various attention strategies.

**Table 1 sensors-24-03090-t001:** Results on TestF under full-reference.

Methods	SHIQ		RD		PSD	
	PSNR (dB)	SSIM	PSNR (dB)	SSIM	PSNR (dB)	SSIM
Tan [44]	11.04	0.4006	12.16	0.4706	13.27	0.49
Yang [45]	14.31	0.5064	13.25	0.4914	14.26	0.48
Shen [24]	13.90	0.4209	12.98	0.5119	20.62	0.8826
Akashi [23]	14.01	0.5267	14.78	0.5726	14.96	0.5524
Shi [27]	18.21	0.6129	19.05	0.6623	18.63	0.6954
Fu [5]	34.13	0.8654	28.19	0.8625	30.02	0.9216
Yi [46]	21.32	0.7216	21.63	0.8002	22.65	0.8442
Huang [47]	28.29	0.8603	27.69	0.8534	31.23	0.9152
Semi-TSGAN	34.69	0.9836	30.45	0.8722	33.27	0.9717

**Table 2 sensors-24-03090-t002:** Results on TestNF under non-reference.

Datasets	Metrics	Methods (Higher, Better)				
		Shi [27]	Fu [5]	Yi [46]	Huang [47]	Semi-TSGAN
SHIQ	DISTS	0.7954	0.8827	0.8413	0.8915	0.9213
	MB-CNN	0.8423	0.9123	0.8754	0.9314	0.9524
	NR-IQA_DmetaLearning	0.6524	0.7264	0.7028	0.7436	0.7542
	MUSIQ	30.16	45.26	39.62	47.19	60.12
RD	DISTS	0.8214	0.8853	0.8621	0.9016	0.9254
	MB-CNN	0.8032	0.8697	0.8316	0.8854	0.9428
	NR-IQA_DmetaLearning	0.6174	0.6745	0.6652	0.7036	0.7292
	MUSIQ	26.54	40.16	35.68	43.69	56.15
PSD	DISTS	0.8023	0.8521	0.8213	0.8897	0.9145
	MB-CNN	0.7995	0.8312	0.8257	0.9036	0.9239
	NR-IQA_DmetaLearning	0.5742	0.6312	0.6069	0.6548	0.6865
	MUSIQ	25.14	43.26	37.61	42.68	48.16

**Table 3 sensors-24-03090-t003:** Ablation studies of diverse configurations in Semi-TSGAN on SHIQ dataset.

Method	SHIQ		RD		PSD	
	PSNR (dB)	SSIM	PSNR (dB)	SSIM	PSNR (dB)	SSIM
Sup-base	29.26	0.9211	25.17	0.8128	28.12	0.8814
Semi-base	27.14	0.8419	23.85	0.7926	25.16	0.8399
Semi-base+DR*	31.86	0.9642	26.56	0.8024	30.16	0.9213
Semi-base+CL*	32.67	0.9625	28.96	0.8319	30.95	0.9126
Semi-TSGAN	34.69	0.9836	30.45	0.8722	33.27	0.9717

**Table 4 sensors-24-03090-t004:** Parameter count and performance metrics for various convolutional layers.

Model	PSNR	SSIM	Parameter Count
FSFA_{*OrdConv*}_	30.35	0.9729	2.15 × 10^5^
FSFA_{3_3_3_3}_	30.45	0.9708	7.76 × 10^4^
FSFA_{1_3_5_7}_	33.17	0.9740	8.99 × 10^4^
FSFA_{5_7_9_11}_	33.73	0.9816	1.39 × 10^5^
FSFA_{3_5_7_9}_	34.69	0.9836	1.10 × 10^5^

**Table 5 sensors-24-03090-t005:** Performance metrics for various fusion strategies.

Fusion Strategy	PSNR	SSIM
Point-wise addition	31.47	0.9750
Matrix concatenation	32.55	0.9782
Only cascade mapping	32.08	0.9762
Only parallel mapping	31.81	0.9782
Cascade & parallel mapping	34.69	0.9836

**Table 6 sensors-24-03090-t006:** Performance metrics for various lightweight modules.

Lightweight Module	PSNR	SSIM	Parameter Count
Shuffle unit	28.99	0.9494	5.25 × 10^4^
Mixconv	29.16	0.9084	1.23 × 10^5^
Xception block	29.81	0.9592	1.64 × 10^5^
FSFA	34.69	0.9836	1.10 × 10^5^

**Table 7 sensors-24-03090-t007:** Performance metrics for various lightweight modules.

Attention Module	PSNR	SSIM	Parameter Count
SE	33.75	0.9793	1.00 × 10^5^
CBAM	33.23	0.9750	1.01 × 10^5^
SA	33.26	0.9777	1.16 × 10^5^
SK	34.16	0.9790	1.31 × 10^5^
OSAtt	34.69	0.9836	1.10 × 10^5^

## Data Availability

The datasets presented in this article are not readily available because the data are part of an ongoing study or due to technical/ time limitations. Requests to access the datasets should be directed to contact corresponding author.

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
