# Peer review of "Semi-TSGAN: Semi-Supervised Learning for Highlight Removal Based on Teacher-Student Generative Adversarial Network"

_sensors, 2024, doi:10.3390/s24103090_

Round 1

Reviewer 1 Report

Comments and Suggestions for Authors

In the manuscript Semi-TSGAN: Semi-Supervised Learning For Highlight Removal based on Teacher-Student Generative Adversarial Network the authors introduce Semi-Supervised Learning based method For image Highlight Removal.

While the paper adequately addresses the utilized framework, evaluation methodology, regularization technique, and architectural enhancements, there are some additional points that should be considered and addressed in detail:

1. A detailed framework needs to be illustrated and explained. Figure 2 does not provide a comprehensive depiction of the framework.

2. Data Dependency: The model's effectiveness heavily relies on the availability and quality of labeled and unlabeled data. Inadequate or biased training data may significantly impact the model's performance and its ability to generalize.

3. Sensitivity to Noise: The model may exhibit sensitivity to noise present in the input data. If the input images contain substantial noise or artifacts, it could negatively affect the accuracy of the highlight removal process.

4. Computational Requirements: The comprehensive feature aggregation module and attention mechanism mentioned in the text may impose significant computational demands. This could limit the model's practicality in resource-constrained environments or on low-power devices.

5. Limited Applicability: The effectiveness of the model could be limited to specific scenarios or datasets. Its performance may vary when applied to different lighting conditions, image resolutions, or various types of highlights. It is crucial to assess the model's robustness and generalization capabilities across a wide range of data.

6. Interpretability: Deep learning models, including the one described, are often regarded as black boxes, making it challenging to interpret and comprehend the decision-making process. This lack of interpretability could be a disadvantage in scenarios where explainability is of utmost importance.

7. Overfitting and Generalization: Despite the utilization of regularization techniques, the model might still be susceptible to overfitting if not appropriately trained or validated. Consequently, it may struggle to generalize well to unseen data.

The above-mentioned points need a thorough analysis and evaluation of the research work to address these potential limitations in a comprehensive manner.

Comments on the Quality of English Language

In the manuscript Semi-TSGAN: Semi-Supervised Learning For Highlight Removal based on Teacher-Student Generative Adversarial Network the authors introduce Semi-Supervised Learning based method For image Highlight Removal.

While the paper adequately addresses the utilized framework, evaluation methodology, regularization technique, and architectural enhancements, there are some additional points that should be considered and addressed in detail:

1. A detailed framework needs to be illustrated and explained. Figure 2 does not provide a comprehensive depiction of the framework.

2. Data Dependency: The model's effectiveness heavily relies on the availability and quality of labeled and unlabeled data. Inadequate or biased training data may significantly impact the model's performance and its ability to generalize.

3. Sensitivity to Noise: The model may exhibit sensitivity to noise present in the input data. If the input images contain substantial noise or artifacts, it could negatively affect the accuracy of the highlight removal process.

4. Computational Requirements: The comprehensive feature aggregation module and attention mechanism mentioned in the text may impose significant computational demands. This could limit the model's practicality in resource-constrained environments or on low-power devices.

5. Limited Applicability: The effectiveness of the model could be limited to specific scenarios or datasets. Its performance may vary when applied to different lighting conditions, image resolutions, or various types of highlights. It is crucial to assess the model's robustness and generalization capabilities across a wide range of data.

6. Interpretability: Deep learning models, including the one described, are often regarded as black boxes, making it challenging to interpret and comprehend the decision-making process. This lack of interpretability could be a disadvantage in scenarios where explainability is of utmost importance.

7. Overfitting and Generalization: Despite the utilization of regularization techniques, the model might still be susceptible to overfitting if not appropriately trained or validated. Consequently, it may struggle to generalize well to unseen data.

The above-mentioned points need a thorough analysis and evaluation of the research work to address these potential limitations in a comprehensive manner.

Author Response

Dear Editors and Reviewers,

We deeply appreciate the time and effort you’ve spent in reviewing our manuscript. Your comments are really thoughtful and helpful. Thus, we revised the manuscript, following your comments exactly.

In the revised manuscript, the modification involves employing the textual content to include the line as a symbolic representation. Here below is our description on revision according to the reviewers’ comments.

 Author's Reply to the Review Report (Reviewer 1)

In the manuscript Semi-TSGAN: Semi-Supervised Learning For Highlight Removal based on Teacher-Student Generative Adversarial Network the authors introduce Semi-Supervised Learning based method For image Highlight Removal. While the paper adequately addresses the utilized framework, evaluation methodology, regularization technique, and architectural enhancements, there are some additional points that should be considered and addressed in detail:

  1. A detailed framework needs to be illustrated and explained. Figure 2 does not provide a comprehensive depiction of the framework.

Reply: Thank you very much. Your suggestion is very valuable for us. We add extra description of Figure2 in order to make it easily sense. To ensure the trustworthiness of pseudo labels assigned to unlabeled data, we establish a dependable repository dedicated to storing top-performing teacher outputs evaluated by NR-IQA. These dependable pseudo labels serve as a reliable compass for guiding the student's training process through the incorporation of unsupervised teacher-student consistency loss  and contrastive loss  mechanisms. The student's weights are iteratively adjusted by minimizing both supervised and unsupervised losses. Furthermore, the teacher model undergoes updates via exponential moving average (EMA) derived from the student's parameters.

  1. Data Dependency: The model's effectiveness heavily relies on the availability and quality of labeled and unlabeled data. Inadequate or biased training data may significantly impact the model's performance and its ability to generalize.

Reply: Thank you very much. Your suggestion is very valuable for us. The quality of the samples is crucial, particularly within the context of semi-supervised learning. This article fails to clarify a fundamental aspect. Nevertheless, regardless of the ratio between labeled and unlabeled samples, appropriate adjustment of the super-re-recessing can be achieved. We add extra description of this concern in Experimental part.

Extra statement: The quality of the samples is crucial, particularly within the context of semi-supervised learning. The ratio of labeled to unlabeled samples varies depending on the dataset, with some datasets exhibiting intrinsic characteristics while others are artificially constructed. Nevertheless, regardless of the ratio between labeled and unlabeled samples, appropriate adjustment of  can be achieved.

  1. Sensitivity to Noise: The model may exhibit sensitivity to noise present in the input data. If the input images contain substantial noise or artifacts, it could negatively affect the accuracy of the highlight removal process.

Reply: Thank you very much. Your suggestion is very valuable for us. The model plays a pivotal role in enhancing the robustness against noise samples. Notably, Semi-TSGAN demonstrates a robust design. A crucial aspect lies in the utilization of contrastive loss, a loss function aimed at enhancing robustness by augmenting the Euclidean distance between dissimilar samples while diminishing the distance between similar ones. Such an approach finds common application in visual authentication domains such as pedestrian recognition. The modified version can be seen in 3.3. Contrastive Regularization For Confirmation Bias Reduction

  1. Computational Requirements: The comprehensive feature aggregation module and attention mechanism mentioned in the text may impose significant computational demands. This could limit the model's practicality in resource-constrained environments or on low-power devices.

Reply: Thank you very much. Your suggestion is very valuable for us. The feature aggregation module can significantly reduce the number of parameters, and the attention mechanism can improve the efficiency of parameter utilization, which increases the memory and computation loss to a certain extent, but this is a balance. Overall, the model is slimmer and performance has increased.

  1. Limited Applicability: The effectiveness of the model could be limited to specific scenarios or datasets. Its performance may vary when applied to different lighting conditions, image resolutions, or various types of highlights. It is crucial to assess the model's robustness and generalization capabilities across a wide range of data.

Reply: Thank you very much. Your suggestion is very valuable for us. Indeed, we used various types of highlights in this paper. The PSD dataset comprises a comprehensive collection of 13,380 images, each captured across 2,210 distinct scenes. Diverse objects and backgrounds are employed to construct this dataset. It is categorized based on two distinct polarization conditions. The first cate-gory encompasses 1,010 sets of images captured at fixed polarization angles, while the second category includes 1,200 image pairs captured with random polarization angles. It can reflect the complexity of highlight formation.

  1. Interpretability: Deep learning models, including the one described, are often regarded as black boxes, making it challenging to interpret and comprehend the decision-making process. This lack of interpretability could be a disadvantage in scenarios where explainability is of utmost importance.

Reply: Thank you very much. Your suggestion is very valuable for us. The interpretability of deep learning is an academic challenge, as is this paper. However, the focus of this article is on highlight suppression in a semi-supervised framework, and the focus is not there. I believe that any study of interpretability will also contribute to our understanding of the properties of Semi-TSGAN.

  1. Overfitting and Generalization: Despite the utilization of regularization techniques, the model might still be susceptible to overfitting if not appropriately trained or validated. Consequently, it may struggle to generalize well to unseen data.

Reply: Thank you very much. Your suggestion is very valuable for us. Semi-supervised learning is essentially a data-driven regularization method, and many mechanisms are added in Semi-TSGAN to ensure that this regularization is positive, such as contrast learning strategies. Our experiments also show that the above directions are helpful for performance improvement.

The above-mentioned points need a thorough analysis and evaluation of the research work to address these potential limitations in a comprehensive manner.

Reviewer 2 Report

Comments and Suggestions for Authors

1. The citation of the image in the third line of the Introduction should be consistent.
2. The symbol is missing at the end of Section 3.3.
3. In the first paragraph of Section 4.3, literature [27] and [5] are mentioned as supervised. What about the others?
4. The comparative methods seem outdated. Please add a comparison with the latest methods.
5. You can consider comparing it with other semi-supervised methods available for image-to-image tasks.

Comments on the Quality of English Language

Minor editing of English language required

Author Response

Dear Editors and Reviewers:

We deeply appreciate the time and effort you’ve spent in reviewing our manuscript. Your comments are really thoughtful and helpful. Thus, we revised the manuscript, following your comments exactly.

In the revised manuscript, the modification involves employing the textual content to include the line as a symbolic representation. Here below is our description on revision according to the reviewers’ comments.

  1. The citation of the image in the third line of the Introduction should be consistent.

Reply: Thank you very much. Your suggestion is very valuable for us. We adjusted the layout of Figure 1.

  1. The symbol is missing at the end of Section 3.3.

Reply: Thank you very much. Your suggestion is very valuable for us. We added missing symbols at the end of Section 3.3.

  1. In the first paragraph of Section 4.3, literature [27] and [5] are mentioned as supervised. What about the others?

Reply: Thank you very much. Your suggestion is very valuable for us. The rest are partly traditional approaches based numerical optimization such as [44][45][23]…. Some are unsupervised Learning such as [47][24] …. The extra emphasis on supervised learning is largely due to the good results it can get, but it also has to do with the approach taken in the original paper.

  1. The comparative methods seem outdated. Please add a comparison with the latest methods.

Reply: Thank you very much. Your suggestion is very valuable for us. It's a little hard to find literature that fits the comparison criteria. To our best knowledge, we are the first ones to introduce a semi-supervised highlight removal framework named Semi-TSGAN. We have referred to several deep learning-based highlight removal comparison papers, and they are the papers with better experimental results on my side.

  1. You can consider comparing it with other semi-supervised methods available for image-to-image tasks.

Reply: Thank you very much. Your suggestion is very valuable for us. Before this, we did a lot of research on relevant papers based on Cycle-GAN, and also did some semi-supervised method experiments, but the effect under the original framework was indeed unsatisfactory, even worse than some customized unsupervised methods, because the addition of comparative experiments was finally abandoned. The reason for personal analysis is that some custom modifications are needed to improve performance.

  1. Comments on the Quality of English Language Minor editing of English language required

Reply: Thank you very much. Your suggestion is very valuable for us. We embellished the whole paper to improve its readability.

Reviewer 3 Report

Comments and Suggestions for Authors

The paper proposes a learning method for removing highlight in images.

I think that the quality of the paper is high and it should be eventually accepted. Authors presented an up-to-date literature overview and their original contribution. The paper also contains experimental results.

My concerns regarding the paper is such that it is quite hard to distinguish in the paper what is Authors original contribution and what is inspired by the work of their researchers. In Section 3 Authors describe the method that they propose but they should include references for parts of the paper which were derived from other papers. I would expect Authors to explicitly present how their network differs from existing networks.

It is also not clear for me what training set did the Authors use. In lines 289-295 they declare preparing their own version of a training set. However, in Sect. 4 they describe sets SHIQ, RD and PSD. Was their method of preparing training sets adapted to all of these benchmarks?

Comments on the Quality of English Language

There are some minor spelling errors in the paper. For example, "Last" instead of "Lastly" in line 19 (abstract). Similarly, in line 104 "To our best knowledge, we introduce". I would suggest: "To our best knowledge, we are the first ones to introduce". However, overall spelling is correct.

Author Response

Dear Editors and Reviewers,

We deeply appreciate the time and effort you’ve spent in reviewing our manuscript. Your comments are really thoughtful and helpful. Thus, we revised the manuscript, following your comments exactly.

In the revised manuscript, the modification involves employing the textual content to include the line as a symbolic representation. Here below is our description on revision according to the reviewers’ comments.

The paper proposes a learning method for removing highlight in images. I think that the quality of the paper is high and it should be eventually accepted. Authors presented an up-to-date literature overview and their original contribution. The paper also contains experimental results.

My concerns regarding the paper is such that it is quite hard to distinguish in the paper what is Authors original contribution and what is inspired by the work of their researchers. In Section 3 Authors describe the method that they propose but they should include references for parts of the paper which were derived from other papers. I would expect Authors to explicitly present how their network differs from existing networks.

Reply: Thank you very much. Your suggestion is very valuable for us. The improvements we propose are in two directions, the first being the innovation of a learning framework, a semi-supervised highlight removal framework based on a teacher-student adversarial generative network model, which is the first time a systematic approach has been proposed for this field. The second is the lightweight adversarial learning network suppression part, the feature aggregation module and the attention module, which belong to the innovative reuse of the new framework. We have further strengthened this idea in the paper to make it easier for readers to read.

It is also not clear for me what training set did the Authors use. In lines 289-295 they declare preparing their own version of a training set. However, in Sect. 4 they describe sets SHIQ, RD and PSD. Was their method of preparing training sets adapted to all of these benchmarks?

Reply: Thank you very much. Your suggestion is very valuable for us. In lines 289-295, we descripted which NR-IQA method is best used to evaluate, using self-produced data sets as the judgment, which is determined before the experiment is used. In Sect 4, we describe sets SHIQ, RD and PSD for experiments using fixed NR-IQA method.

There are some minor spelling errors in the paper. For example, "Last" instead of "Lastly" in line 19 (abstract). Similarly, in line 104 "To our best knowledge, we introduce". I would suggest: "To our best knowledge, we are the first ones to introduce". However, overall spelling is correct.

Reply: Thank you very much. Your suggestion is very valuable for us. We have modified it as required.

Round 2

Reviewer 1 Report

Comments and Suggestions for Authors

The first review comments were addressed properly. Thank you.  

Reviewer 3 Report

Comments and Suggestions for Authors

I have no further comments to the paper.